# Half a degree of warming may cause double the economic loss and increase the population affected by floods in China

Lulu Liu [1], Jiangbo Gao[1], Shanghong Wu[1,2]

[1]Key Laboratory of Land Surface Pattern and Simulation, Institute of Geographic Sciences and Natural Resources Research, Chinese Academy of Sciences, Beijing, 100101, China
[2]University of Chinese Academy of Sciences, Beijing, 100049, China

Correspondence: Jiangbo Gao (gaojiangbo@igsnrr.ac.cn)

**Abstract.** Based on future scenario data and an improved quantitative natural disaster risk assessment model, in this study, we analysed the response of the characteristics of flood events in China to 1.5°C and 2°C of global warming, quantitatively assessed the population affected and the economic risks of floods, and determined the integrated risk levels. The results indicate that for RCP4.5 and RCP8.5 scenarios, the probability and distribution area of the floods increase with increasing temperature and the influence range of the floods of different levels expands more rapidly under RCP4.5 scenario. The floods mainly affect the social economy in the regions with lower altitudes and smaller slopes in eastern China. As the increase in temperature intensifies, the population affected and the direct economic losses are aggravated. For 2°C of global warming, under RCP8.5 scenario, the population affected by floods increases by 2 million and the economic risk nearly doubles compared with 1.5°C of global warming. The economic risk under RCP4.5 scenario even reaches three times that for 1.5°C of global warming, but its proportion to the gross domestic product (GDP) is lower than that under RCP8.5 scenario. Under both scenarios, the ranges of the medium-high flood risk zones gradually expand westward and northward.

## 1 Introduction

In recent decades, climate changes have affected natural and human systems on all of the continents and across the oceans (IPCC, 2012; IPCC, 2014). At the global and regional scales, climate-related hazards have caused enormous damage (Kundzewicz et al., 2014; Johnson et al., 2016; Luo et al., 2018; Paprotny et al., 2018), resulting in a significant increase in economic losses (Parmesan and Yohe, 2003; Patz et al., 2005). Between 2000 and 2019, 7348 disaster events were recorded worldwide, resulting in 1.23 million deaths, $2.97 trillion USD in economic losses and over 4 billion people affected, with a surge in the number of climate-related disasters, of which floods were the most frequent, accounting for 44% of all disasters. (CRED and UNDRR, 2020; WMO, 2021). Over the last 20 years, China has suffered the second largest loss due to the frequent occurrence and serious impacts of climate-related hazards (Ding et al., 2006; Huang et al., 2007; UNISDR and CRED, 2018). Among all of the meteorological and hydrological disasters, the floods were characterized by sudden and frequent occurrences, and the direct economic losses caused by floods accounted for the largest proportion of the total economic losses due to climate-

related hazards (Writing Committee for Third National Assessment Report on Climate Change, 2015). Over the last ten years, the mean annual direct economic loss due to floods in China has exceeded $25 billion USD, and the affected population has exceeded 100 million with more than 1000 deaths (Song, 2019; Chen et al., 2022).

In order to reduce the risks and impacts of climate change and to strengthen the global response to the threats posed by climate change, the Paris Agreement committed to controlling the increase in the global mean surface temperature (GMST) to "well below 2°C above preindustrial levels and pursuing efforts to limit the temperature increase to 1.5°C above pre-industrial levels" (UNFCCC, 2015). The Intergovernmental Panel on Climate Change (IPCC) Special Report on the impacts of 1.5°C of global warming states that 1.5°C of global warming can effectively reduce the risks posed by climate change and avoid the occurrence of irreversible risks and losses (IPCC, 2018). The IPCC's Sixth Assessment Report indicates that global surface

temperature was 1.09°C higher in 2011–2020 than 1850–1900 and that global warming will reach 1.5°C in the near-term, which will cause unavoidable increases in multiple climate hazards and present multiple risks to ecosystems and humans (IPCC, 2021; IPCC, 2022). Globally, climate change is expected to accelerate the global hydrologic cycle (Held and Soden, 2006; Huntington, 2006; Durack et al., 2012; Trenberth et al., 2014), and increase the frequency and intensity of extreme rainfall events and severe storm surges (Wahl et al., 2015; King et al., 2017a, b; Rahmstorf, 2017; Kharin et al., 2018; Li et al., 2018b;

Zhang et al., 2018). Upper middle income countries will suffer the largest increase in flood damage, and China will suffer the most serious direct economic losses (Jevrejeva et al., 2018; Willner et al., 2018). The regions with high flood risk levels are mainly located in southeastern China (Xu et al., 2014). The frequency of events similar to the extreme floods during the summer of 2010 will increase by two and three times for global warming of 1.5°C and 2°C, respectively (Lin et al., 2018). The socioeconomic exposure will mainly increase in the economically developed regions in eastern China (Li et al., 2018a).

Previous studies have paid more attention to the possibility of floods (Hirabayashi et al., 2013; Lin et al., 2018) and have determined the socioeconomic risks of floods using simple superposition of the risk causing factors and exposures (Li et al., 2018). Few studies have been conducted quantitative assessments of flood risks that consider the vulnerability of the risk bearing bodies (Alfieri et al., 2015; Wobus et al., 2017). However, the connotation of risk should include the risk causing factors, disaster-pregnant environments and risk bearing bodies. The adverse effects of floods depend not only on the extreme

events themselves, but also on the exposure and vulnerability of the risk bearing bodies (IPCC, 2012). At present, it is urgent to quantitatively assess the population affected and the economic risks of floods and their trends under different global warming targets by clarifying the magnitude and frequency of floods, the vulnerability, and the social and economic exposure in order to improve the analysis of the risk level, reduce losses caused by floods, and provide a basis for addressing climate change and preventing natural disasters under different global warming targets.

Based on the quantitative evaluation theory for climate change risk and the magnitude and frequency of pluvial floods, in this study, we calculated the probability of floods in China for 1.5°C and 2.0°C of global warming using the bias-corrected daily output results of global climate models. By combining the socioeconomic exposure and the latest vulnerability to floods of different levels, the population affected and the economic risks posed by floods were quantitatively assessed to provide

support for establishing disaster management systems, carrying out effective risk management, ensuring sustainable development of the national economy and society, and implementing adaptation and mitigation strategies.

## 2 Materials and Methods

### 2.1 Data sources

We selected five Global Climate Models (HadGEM2-ES, GFDL-ESM2M, IPSL-CM5A-LR, MIROC-ESMCHEM, and NorESM1-M) from the Inter-Sectoral Impact Model Inter-Comparison Project (ISI-MIP) driven by multiple Representative Concentration Pathways (RCPs) scenarios (Warszawski et al., 2014). The climate projections were bias-corrected using method of modifying the transfer function based on water and global change (WATCH) forcing data (WFD) (Weedon et al., 2011; Hempel et al., 2013) with a spatial resolution of 0.5°×0.5° and a daily step from 1950 to 2099. Researches show that global warming of 1.5°C is becoming inevitable. Therefore, two different scenarios, RCP4.5 and RCP8.5, which represent moderate development path and high emission path respectively, were selected in this study to reflect the spatial and temporal patterns of risk under the possible development path of current policies and under the path of no response measures. The simulation results satisfactorily represent the ranges of the climate change and the corresponding impacts of the other CMIP5 models (Huang et al., 2017). Thus, they were used to calculate the magnitude and frequency of floods and to classify their hazards.

Social and economic data derived from the Shared Socioeconomic Pathways (SSPs) scenarios (O'Neill et al., 2014) and from downscaling of the scenario datasets, including the population and gross domestic product (GDP), were used by the National Institute for Environmental Studies, Japan to create a simulation based on the SSP Database of the International Institute for Applied Systems Analysis (IIASA). The data interval was 10 years from 1980 to 2100, and the spatial resolution was 0.5°×0.5° (Murakami and Yamagata, 2016). The SSP1 and SSP3 scenarios were selected, which correspond to RCP4.5 and RCP8.5 scenarios, respectively (Kriegler et al., 2010; Van Vuuren et al., 2012). The data were used to assess the population affected and the economic risks posed by floods to represent the exposure of the risk bearing bodies.

### 2.2 Analysis methods

### 2.2.1 Determination of 1.5°C and 2°C global warming periods

We selected global warming periods of 1.5°C and 2°C based on the objectives of the Paris Agreement. Previous studies found that the GMST is projected to reach 1.5°C by around 2030 for both RCPs whereas the 2°C is reached by 2040 under RCP8.5 and by 2050 under RCP4.5 (Karmalkar and Bradley, 2017; Nikulin et al., 2018; Su et al., 2018; Gao et al., 2020). We chose 30 years as the threshold year and the last year as the warming period for 1.5°C (2.0°C).

## 2.2.2 Quantitative natural disaster risk assessment model

The quantitative natural disaster risk assessment model consists of three components: the destructive power of the natural disasters or the damage standard of the risk bearing bodies (D), the exposure of the risk bearing bodies (E), and the probability of disaster or disaster-pregnant environment (P) (Wu et al., 2018a). These three components were combined with the most commonly used equal weighting method (Papathoma-Köhle et al., 2019):

$$R = D \times E \times P \tag{1}$$

Bates et al. (2008) pointed out that floods are affected by a variety of climatic and non-climatic processes. Precipitation is the most important climatic process. Flood disasters in China are mostly caused by intense precipitation, and the intensity of and losses caused by the floods are closely related to the corresponding precipitation. Therefore, the amount of precipitation can be used to classify the intensity of floods (Zhai et al., 1999; Gong and Wang, 2000; Zhai et al., 2005; Ma et al., 2018). From the perspective of historical flood disasters, the maximum accumulated 3-day precipitation has an important impact on the occurrence of floods during periods of continuous precipitation. Generally, mild, moderate, and severe floods correspond to maximum accumulated 3-day precipitation values of 30(35)–150 mm, 150–250 mm, and ≥ 250 mm, respectively (Li et al., 2012). For flood disasters in the south of the Yangtze River, the maximum accumulated 3-day precipitation of more than 35 mm is generally required, whereas for flood disasters in the north of the Yangtze River, the maximum accumulated 3-day precipitation of more than 30 mm is sufficient. Therefore, for the classification of mild flood disasters, the maximum accumulated 3-day precipitation is at a minimum of 35 mm in the south of the Yangtze River and 30 mm in the north of the Yangtze River. Among the non-climatic processes, the altitude and slope have the greatest impacts on the formation and development of floods, so they can be used as environmental correction parameters for the underlying surface (Zhou et al., 2000; Thompson and Clayton, 2002; Li et al., 2012).

Based on the above theories, the probability of floods can be expressed as follows:

$$P = F \times I \tag{2}$$

where $P$ is the probability of flood and can be used to characterise its hazard, $F$ is the possibility of intense precipitation, and $I$ is the environmental correction parameter for the underlying surface. The process of constructing the environmental correction parameter for the underlying surface was as follows: firstly, the different watersheds are divided, and then the elevation and slope corresponding to the approximate affected area of historical mild, moderate and severe floods in the watershed were analysed to derive the grading criteria for elevations and slopes, and finally the environmental correction parameter for the underlying surface were obtained using the equal weighting method (Li et al., 2012).

Correspondingly, the quantitative assessment model of the population affected and the economic risks posed by pluvial floods can be expressed as follows:

$$R = (D \times E) \times (F \times I) \tag{3}$$

where $R$ is the population affected and the economic risks posed by the floods, $D$ is the destructive power of the floods with different magnitudes, and $E$ is the exposure of the risk bearing bodies. The destructive power of the floods is obtained based

on the relationship between flood hazards of different intensity and social economic losses. Firstly, the vulnerability curves of the maximum accumulated 3-day precipitation and the affected population rate or direct economic loss rate during historical flood disasters were constructed in different regions. Finally, the loss criteria for the population affected and the direct economic losses posed by the floods in each region were established (Li et al., 2012).

Therefore, the steps of the integrated risk assessment of pluvial floods in China are as follows:

The maximum accumulated 3-day precipitation values of 30(35)–150 mm, 150–250 mm, and ≥ 250 mm were taken as the maximum occurrence of mild, moderate, and severe floods. These values were converted into a probability (probability = frequency/time period × 100%, set to 1 if the probability is greater than 100%). Furthermore, considering the heterogeneity of the regional geographical environment, the environmental correction parameter of the underlying surface, which is based on the altitude and slope, was used to correct this probability. Thus, the hazards of the floods were obtained;

Flood disasters can have serious impacts on the social economy. The GDP and population were selected to represent the exposure of the social economy. For the destructive power of floods, a quantitative relationship was established between the floods of different magnitudes and the loss rates of the GDP and population according to the classification criterion of the maximum accumulated 3-day precipitation based on statistical data for 1001 flood disasters from 1990 to 2008 in China, which was obtained from Li et al. (2012). The spatial distribution patterns of the affected population rates and direct economic loss rates for different levels of floods were shown in Fig. S1;

The exposures of the social economy were obtained from the SSPs data described in the data sources section. The annual population and GDP data were obtained based on linear interpolation method, and then the average values of population and GDP at different global warming periods under different scenarios were calculated to represent the exposure (Fig. S2; Fig. S3);

The population affected and the economic risks posed by floods were quantitatively evaluated based on the hazards posed by and destructive power of floods and the exposures of the risk bearing bodies;

Finally, the integrated risk of the floods was obtained using the superposition analysis method and was graded using the multiple of the standard deviation method.

## 3 Results

### 3.1 The probability of floods

For a 1.5°C increase in the GMST, under RCP8.5 scenario, the distribution areas of the severe, moderate, and mild floods would about 1.92, 3.90, and 6.92 million km$^2$, respectively (Fig. 1). Generally speaking, the severe floods would mainly be concentrated in South China, East China, the southeastern part of the Qinghai-Tibet Plateau, the southern part of Northeast China, and the southwestern part of Southwest China (Fig. 2a). The moderate floods would be distributed in South China, East China, Central China, the central and southern parts of Northeast China, the southern part of North China, and the southern part of Southwest China (Fig. 2c). The mild floods would occur in most parts of the country, except on the Qinghai-Tibet Plateau and in the Tarim Basin (Fig. 2e). Under RCP4.5 scenario, the probability of severe floods in Central China would be

higher, and the distribution areas of the severe, moderate, and mild floods would be larger than those under RCP8.5 scenario (Fig. S4a, c, e). For a 2°C increase in the GMST, under RCP8.5 scenario, the distribution areas of the severe and moderate floods would increase to approximately 2.09 and 4.12 million km$^2$, respectively (Fig. 1). The probability of severe and moderate floods would increase in the eastern part of Central China (Fig. 2b, d). The distribution area and probability of mild floods would be basically the same as that for 1.5°C of global warming (Fig. 2f). Under RCP4.5 scenario, the distribution areas of the floods of different levels would increase, the probability of severe floods in the central and northern parts of East China would increase, and the moderate and mild flood zones would expand westward (Fig. S4b, d, f).

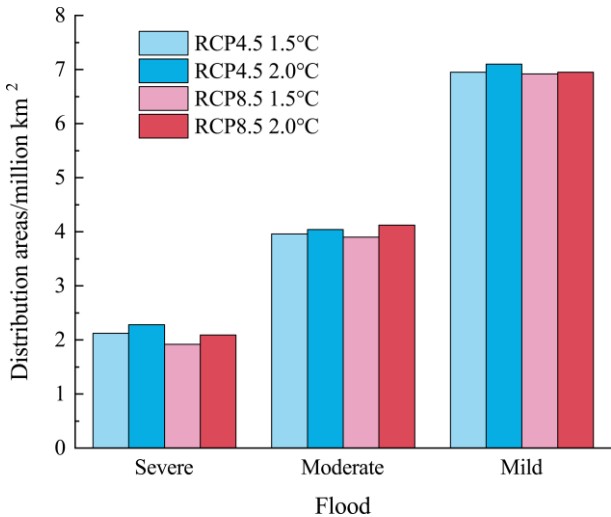

**Figure 1.** Distribution areas of the severe, moderate, and mild floods for 1.5°C and 2°C of global warming (units: million km$^2$)

Compared with 1.5°C of global warming, the probability of the severe flood would be significantly higher for global warming of 2°C under RCP4.5 and RCP8.5 scenarios, and in some areas, the probability would double. The probability would decrease in the coastal region of East China, Central China and South China under RCP4.5 scenario. In contrast, under RCP8.5 scenario, the probability would only decrease in the southern part of Central China and the southern part of Yunnan Province, and it would increase significantly in South China and the northern part of Central China (Fig. S5a, b). The probability of the moderate flood would double on the North China Plain and the Yunnan-Guizhou Plateau (the eastern part of Northeast China) under RCP4.5 (RCP8.5) scenario, and the probability would decrease in the Northeast Plain, Loess Plateau, and the southern part of Central China under both scenarios (Fig. S5c, d). The changes in the mild hazard index would mainly be concentrated in the Northwest China, with an increase in the eastern region and a decrease in the western region under RCP4.5 and RCP8.5 scenarios (Fig. S5e, f).

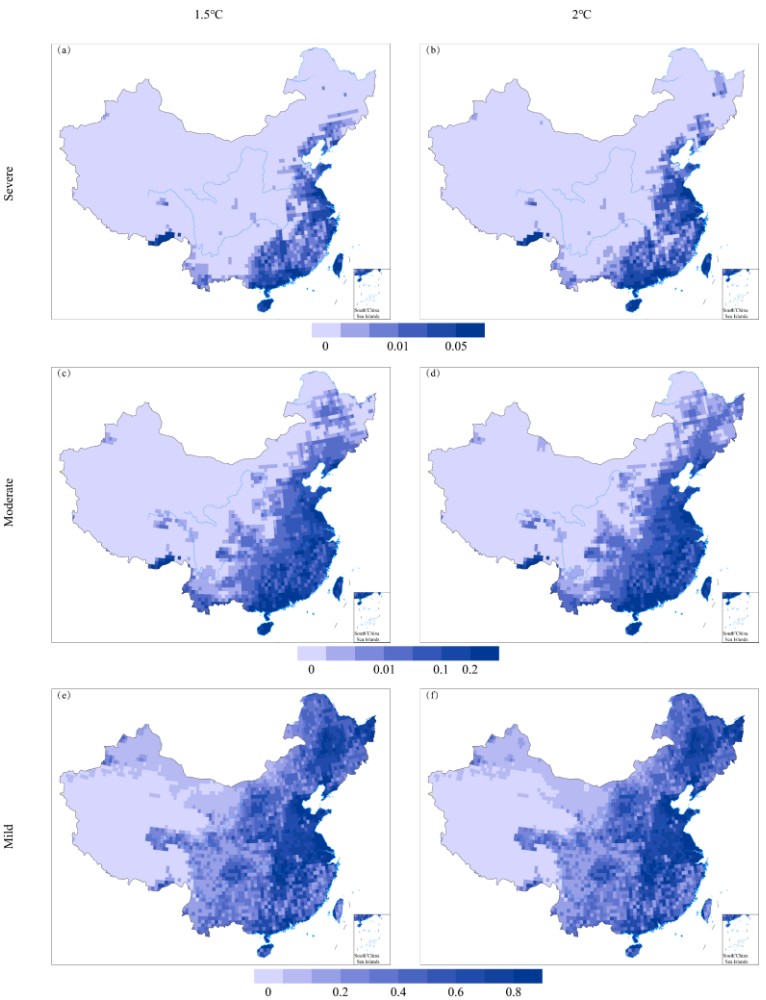

**Figure 2.** Spatial patterns of the probability of the severe **(a-b)**, moderate **(c-d)**, and mild **(e-f)** floods for 1.5°C **(a, c, e)** and 2°C **(b, d, f)** of global warming under RCP8.5 scenario

## 3.2 The population affected by floods

Floods have serious impacts and cause damage to the social economy, food production, natural ecosystem, and infrastructure. In this study, the population and economy were selected as the risk bearing bodies in order to quantitatively assess the risks posed by floods in the future.

For 1.5°C of global warming, under RCP8.5 scenario, population affected by the severe floods would exceed 11 million people, with the most population affected mainly distributed in the coastal areas of East China, the coastal areas of southern China and southern Tibet, followed by the southern part of Northeast China, the eastern part of North China, the central part of East China, the southern part of Central China and the northern part of South China (Fig. 3; Fig. 4a). The moderate floods

would affect about 26 million people, and the high risks would be mainly concentrated in the Bohai Rim, the central and northern parts of East China, the southern part of South China, Central China, followed by the eastern part of Southwest China and southern Tibet (Fig. 3; Fig. 4c). The population affected by mild floods would be about 55 million, with the regions with high risk levels mainly in the eastern part of North China, the coastal areas of East China and the coastal areas of South China, followed by the eastern part of Northwest China, the eastern part of Southwest China, the northern part of Xinjiang and southern Tibet (Fig. 3; Fig. 4e). The total number of affected populations under RCP4.5 scenario would be slightly lower than that under RCP8.5 scenario, and the spatial distribution would be basically consistent with that under RCP8.5 scenario, with the scope would shift westward (Fig. 3; Fig. S6a, c, e). For 2°C of global warming, under RCP8.5 scenario, compared with 1.5°C of global warming, the population affected by severe floods would increase by about 2 million, and the impact range would expand northward and westward (Fig. 3; Fig. 4b). The moderate floods would affect nearly 29 million people, and the risk in Central China would increase (Fig. 3; Fig. 4d). The population affected by mild floods would increase by more than 2 million, but the impact range would be basically the same as that for 1.5°C of global warming (Fig. 3; Fig. 4f). Under RCP4.5 scenario, the population affected by floods of different levels would be lower. The impact ranges of the two scenarios would be basically the same (Fig. 3; Fig. S6b, d, f).

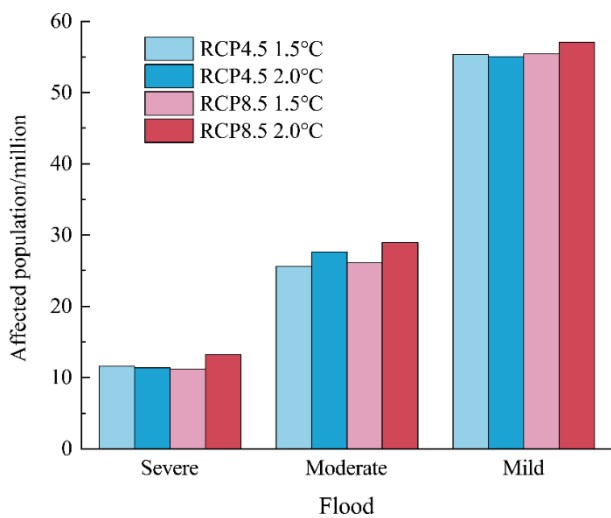

**Figure 3.** The population affected by the severe, moderate, and mild floods for 1.5°C and 2°C of global warming (units: million)

Compared with 1.5°C of global warming, the risks to population posed by severe floods under RCP8.5 scenario for 2°C of global warming would increase significantly in most of eastern China, except for the southern part of Northeast China, the southern part of Central China, the southern part of Yunnan and Taiwan province; and in some areas the risk would double (Fig. S7a). The risks to population of severe floods under RCP4.5 scenario would increase in the southern part of Northeast China, East China and the southern part of Southwest China, and they would decrease in Central China and South China (Fig. S7b). The changes in the risks to population posed by moderate floods would mainly be distributed to the east of the Heihe-

Tengchong Line. The distribution patterns would be similar under both scenarios, and the areas of increased risk would mainly be concentrated in the North China Plain and the Yangtze River Basin (Fig. S7c, d). The change in the risks to population posed by mild floods to the east of the Heihe-Tengchong Line would be small, while the change to the west of the Heihe-Tengchong Line would be large, with the most significant increases mainly occurring in the northern part of North China, Northwest China and Sanjiangyuan (Fig. S7e, f).

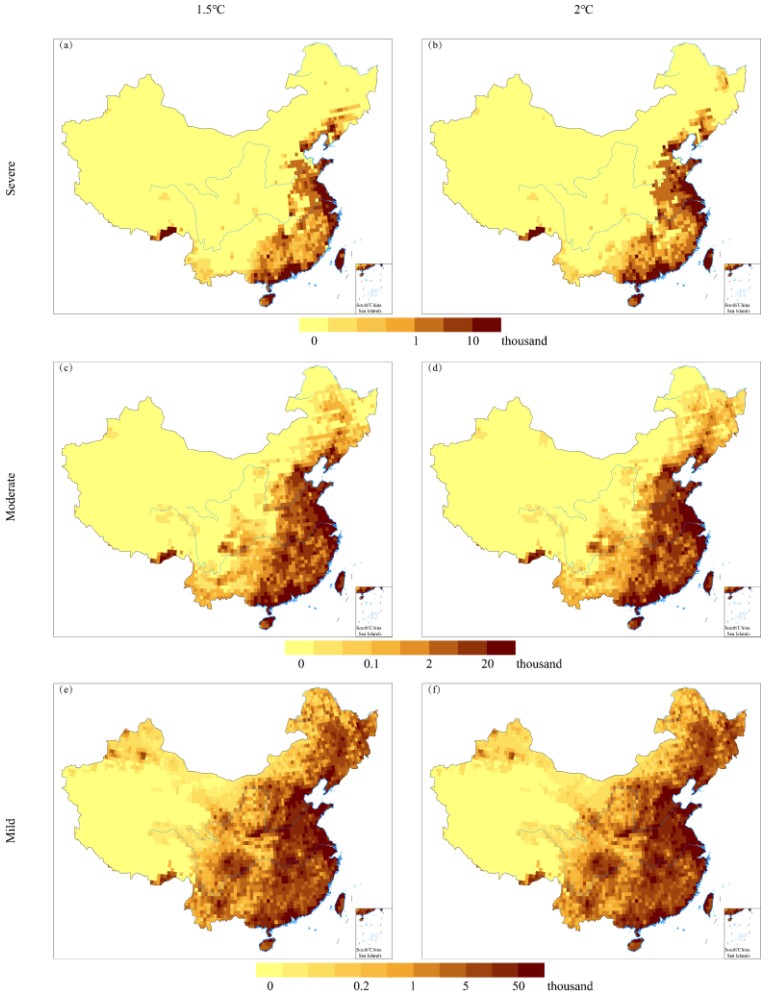

**Figure 4.** Spatial patterns of the population affected by the severe **(a-b)**, moderate **(c-d)**, and mild **(e-f)** floods for 1.5°C **(a, c, e)** and 2°C **(b, d, f)** of global warming under RCP8.5 scenario

## 3.3 The economic risks posed by floods

For 1.5°C of global warming, under RCP8.5 scenario, the direct economic losses posed by the severe floods would be about $33 billion USD (Fig. 5). The regions with the highest direct economic losses were mainly in the Yangtze-Huaihe region,

the Pearl River basin and southern Tibet, followed by the Jiangnan region (Fig. 6a). The moderate floods would cause direct economic losses of approximately $42 billion USD, with high risk occurring mainly in the North China Plain, the middle and lower reaches of the Yangtze River, the Pearl River basin and Taiwan province, followed by the Sichuan Basin and southern Tibet (Fig. 5; Fig. 6c). The direct economic losses posed by mild floods would be about $70 billion USD, and the regions with
230   high risk levels would mainly be distributed in the North China Plain, the Northeast Plain, the Middle-Lower Yangtze Plain, the Sichuan Basin, and the Pearl River Basin (Fig. 5; Fig. 6e). The total direct economic losses and their spatial distribution under RCP4.5 scenario would be basically consistent with those under RCP8.5 scenario (Fig. 5; Fig. S8a, c, e). For 2°C of global warming, under RCP8.5 scenario, compared with 1.5°C of global warming, the direct economic losses posed by severe floods would double, and the risk in eastern China would increase (Fig. 5; Fig. 6b). The moderate floods would cause direct
235   economic losses approach $78 billion USD, with economic risk increasing significantly in the plains of eastern China (Fig. 5; Fig. 6d). The direct economic losses posed by mild floods would nearly double, and the impact range would be basically the same as that for 1.5°C of global warming, with economic risk increasing significantly in the North China Plain (Fig. 5; Fig. 6f). Under RCP4.5 scenario, the direct economic losses posed by floods of different levels would be significantly higher. However, the proportion of the GDP would be less than that under RCP8.5 scenario (Fig. 5; Fig. S8b, d, f).

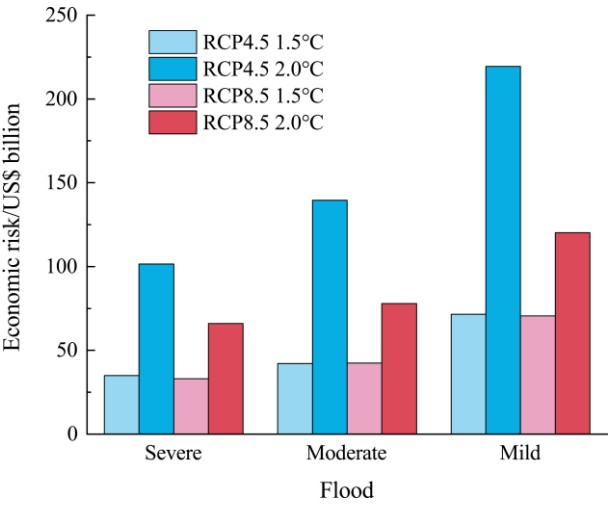

240
**Figure 5.** Economic risks of floods of different levels for 1.5°C and 2°C of global warming (units: US$ billion)

Compared with 1.5°C of global warming, the economic risk posed by floods of different grades would mainly increase under the effect of economic growth for 2°C of global warming. The economic risks posed by severe floods under RCP4.5 and
245   RCP8.5 scenarios would decrease in parts of Central China and would increase in most of eastern China, and the risk in the North China Plain would increase by more than 500% (Fig. S9a, b). Under RCP8.5 scenario, the economic risks posed by moderate floods would decrease on both sides of the Heihe-Tengchong Line and would nearly double on the eastern side of the Heihe-Tengchong Line, with the risk increasing more than fivefold in the Northeast Plain and the Qinling Mountains (Fig. S9c). Under RCP4.5 scenario, the risks would decrease on both sides of the Heihe-Tengchong Line and in the eastern part of Northeast

China. The risks in most areas to the east of the Heihe-Tengchong Line would increase by about twofold, and the risks in the southern part of Southwest China would increase by more than 10 times (Fig. S9d). Under RCP8.5 scenario, the economic risks posed by mild floods would increase within 100% in most parts of the country, and the risk would increase by more than two times in the northern part of Northwest China (Fig. S9e). Under RCP4.5 scenario, the risks in most parts of the country would increase by more than 200%, and the risk in the northern part of Northwest China and in Sanjiangyuan would increase by more than 10 times (Fig. S9f).

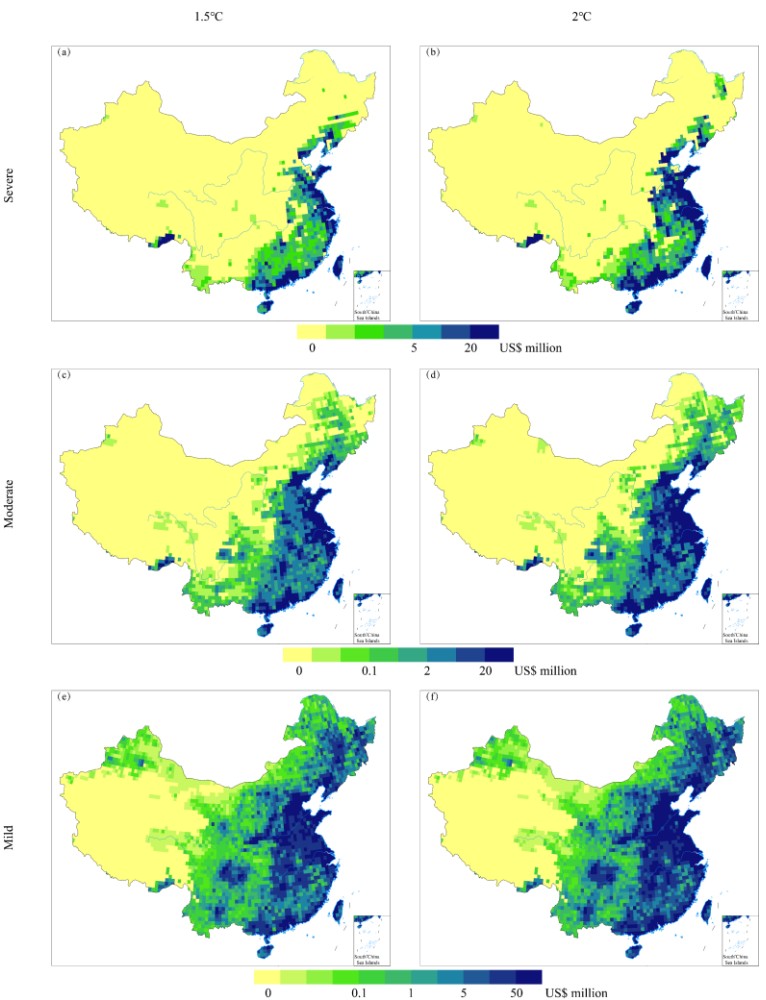

**Figure 6.** Spatial patterns of the economic risks posed by the severe **(a-b)**, moderate **(c-d)**, and mild **(e-f)** floods for 1.5°C **(a, c, e)** and 2°C **(b, d, f)** of global warming under RCP8.5 scenario

## 3.4 Integrated risks posed by floods

Based on the above quantitative assessment of the risks to the population and economy posed by floods, the standard deviation method was used to assess the integrated risks posed by floods.

For 1.5°C of global warming, the risk distributions under RCP4.5 and RCP8.5 scenarios would be generally consistent. The integrated risks posed by floods would occur in most parts of the country, except on the Qinghai-Tibet Plateau and in the Tarim Basin. The high level risk regions would mainly be concentrated in East China, South China, Central China, the central part of Northeast China, the eastern part of North China and southern Tibet. The medium level risk regions would mainly be distributed in Northeast China, the central part of North China, the eastern part of Northwest China, the central and western parts of Southwest China. The medium-high risk zones would account for 40% of the national total land area of China (Fig. 7a, c). For 2°C of global warming, under RCP8.5 scenario, the distribution areas of the high and medium risk zones would increase to different extents, while the distribution areas of the low risk zones would decrease. The expansion of the medium risk zone would be the most significant, and would mainly affect the eastern part of Northwest China. The medium-high risk zones would account for about 45% of the total land area in China. Under RCP4.5 scenario, the distribution areas of the high risk and medium risk zones would increase slightly, mainly in Southwest China and the eastern part of Northwest China, while the distribution area of the low risk zones would decrease slightly (Fig. 7b, d).

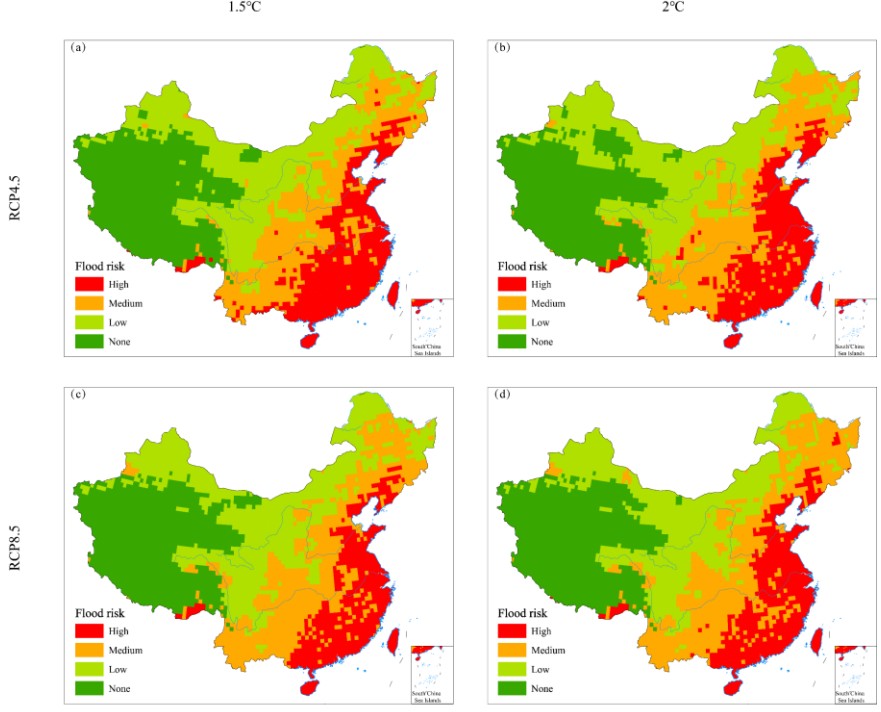

**Figure 7.** Spatial patterns of the integrated risks posed by floods for 1.5°C **(a, c)** and 2°C **(b, d)** of global warming under RCP4.5 **(a, b)** and RCP8.5 **(c, d)** scenarios

## 4 Discussion

In the future, under the continuous intensification of global warming, the impact and distribution area of severe and moderate floods with more serious impacts would continue to expand, and the population affected and economic risks posed by floods would become more serious. In terms of climate scenarios, the distribution area of the floods under RCP4.5 scenario would be broader; however, under RCP8.5 scenario, the floods would affect more people, the direct economic losses would account for a larger proportion of the GDP, and the adverse impacts on the social economy would be more serious. Regarding the

contributions of the various components to the floods risks, we discuss the severe floods in Northeast China (including Heilongjiang, Jilin, Liaoning, and the eastern part of Inner Mongolia) as an example. According to Equation (3), for 1.5°C of global warming, the population affected (units: million) and the direct economic losses (units: US\$ billion) caused by severe floods in Northeast China would be $0.2880 \approx 30.73\% \times 129.8462 \times 0.0130 \times 0.5552$ and $1.0518 \approx 11.83\% \times 1231.8067 \times 0.0130 \times 0.5552$, respectively. For 2°C of global warming, the population affected and the direct economic losses caused by

severe floods would be $0.3615 \approx 30.73\% \times 145.1178 \times 0.0146 \times 0.5552$ and $2.4794 \approx 11.83\% \times 2585.6269 \times 0.0146 \times 0.5552$, respectively. Because the vulnerability and the environmental correction parameter would be basically unchanged, the increase in the risk would mainly depend on the hazards and exposure. The contribution of the hazards and exposure to the population risk would be about the same, and the contribution of the exposure to the economic risk would be about twice that of the hazards. By understanding floods and their impacts, it is urgent to clarify the required adaption to climate change and

implement plans for the different possible situations in order to reduce disaster losses and ensure the sustainable development of the social economy (Jevrejeva et al., 2018; Jongman, 2018; Lim et al., 2018; Wu et al., 2018b).

Future intensification of global warming (IPCC, 2013) and the frequent occurrence of extreme meteorological and hydrological events (Rahmstorf and Coumou, 2011; Bao et al., 2017; Kendon, 2018) will have an important impact on the social economy, natural ecosystem, and food production (Piao et al., 2010; Bellard et al., 2012; Burke et al., 2015; Carleton

and Hsiang, 2016; Lesk et al., 2016). The losses caused by floods are closely related to the socio-economic background. The exposure and vulnerability of risk bearing bodies will change in response to climate change, social and economic development, and disaster prevention and mitigation capabilities. The uncertainty of the quantitative assessment of the social and economic risk posed by pluvial floods includes three components (i.e., the probability or hazard, the vulnerability, and the exposure). There is uncertainty in the simulation and downscaling of global climate models (GCMs). The use of multi-model ensemble

methods and the improvement of the simulation accuracy are conducive to the reducing of uncertainty. The provincial scale vulnerability of the population affected and the direct economic losses is insufficient to characterize the spatial heterogeneity. Future research needs to continuously introduce the latest disaster data, modify the vulnerability curve, and improve the reliability of the vulnerability assessment (Huang et al., 2012; Ouma and Tateishi, 2014). It is difficult to reflect the impact of climate change and climate policies using exposure data from SSPs, which leads to a systematic deviation in the risk assessment.

It is necessary to build a high spatial-temporal resolution SSP basic element dataset combined with China's national conditions (Chen et al., 2020).

In this study, the population affected and the economic risks posed by precipitation-generated inland floods were examined, but coastal floods caused by storm surges and sea level rise were not considered. The factors affecting the formation of inland floods also include the antecedent conditions of rivers and their drainage basins, the properties and status of the soil, the

vegetation cover, and urbanization. This study mainly focused on large-scale flood disasters at the national level. The influence of the antecedent conditions of rivers and their drainage basins is reflected in the altitude and slope. Factors such as soil, vegetation, and urbanization mainly affect the formation of small-scale floods (Blöschl et al., 2007). Therefore, the environmental correction parameter for the underlying surface does not consider the above factors. Of course, when studying small-scale or regional flood disasters, the above factors should be considered according to the actual situation (Kron, 2005;

Bates et al., 2008; Kundzewicz et al., 2014).

**5 Conclusions**

Based on future scenario data, in this study, we calculated the probability of floods under 1.5°C and 2°C of global warming and quantitatively assessed the population affected and the economic risks posed by floods. The following conclusions were drawn.

(1) In the future, the probability and impact range of floods of different levels in China would increase, and the area affected by severe floods would expand the fastest. For 2°C of global warming, severe floods would cover more than 2 million km$^2$ under RCP4.5 and RCP8.5 scenarios.

(2) Floods mainly affect the population and economy of eastern China. The population affected by and the direct economic loss caused by floods would continue to increase with increasing global warming under different climatic and socioeconomic

scenarios. Compared to 1.5°C of global warming, for 2°C of global warming, the population affected by severe floods would increase by about 2 million under RCP8.5 scenario and would decrease slightly under RCP4.5 scenario. The economic risk would double under RCP8.5 scenario and would triple under RCP4.5 scenario, but the proportion of the GDP would be less than that for RCP8.5 scenario.

(3) The regions with medium-high integrated risks would mainly be distributed in Northeast China, North China, East China,

Central China, and South China. With increasing climate change and socioeconomic development, the range of the medium-high flood risk zones would gradually expand westward and northward. For 2°C of global warming, the area of the medium-high risk zones would account for about half of the total land area in China under RCP8.5 scenario.

*Data availability*. The climate scenario data are provided by the Inter-Sectoral Impact Model Inter-Comparison Project (ISI-

MIP). The socioeconomic scenario data are provided by the National Institute for Environmental Studies, Japan.

*Supplement*. The supplement related to this article is available online at: https://doi.org/10.5194/NHESS-2021-304-supplement.

*Author contributions*. LL and JG designed the research framework and developed the methodology. LL prepared the original draft. JG guided the research process and revised the manuscript following reviewers' suggestions. SW supervised the project and provided advice and feedback in the process. All authors discussed the results and contributed to the final version of the paper.

*Competing interests*. The contact author has declared that neither they nor their co-authors have any competing interests.

*Disclaimer*. Publisher's note: Copernicus Publications remains neutral with regard to jurisdictional claims in published maps and institutional affiliations.

*Special issue statement*. This article is part of the special issue "Advances in flood forecasting and early warning". It is not associated with a conference.

*Acknowledgements*. This work was mainly supported by the National Key Research and Development Program of China (grant no. 2018YFC1508900) and the Strategic Priority Research Program of the Chinese Academy of Sciences (grant no. XDA19040304) and the National Natural Science Foundation of China (grant no. 42101311) and the China Postdoctoral Science Foundation (grant no. 2021M670433). The authors would like to acknowledge Jie Yin and the three anonymous referees for the thoughtful comments.

*Financial support*. This research has been supported by the National Key Research and Development Program of China (grant no. 2018YFC1508900), the Strategic Priority Research Program of the Chinese Academy of Sciences (grant no. XDA19040304), the National Natural Science Foundation of China (grant no. 42101311), and the China Postdoctoral Science Foundation (grant no. 2021M670433).

*Review statement*. This paper was edited by Jie Yin and reviewed by two anonymous referees.

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
