# Peer review of "Half a degree of warming may cause double the economic loss and increase the population affected by floods in China"

_Natural Hazards and Earth System Sciences, 2021_

## Author Response (AR1)

Review 1:

Flood risk assessment is an important but difficult issue especially under the global change. This manuscript assessed the flood risk of economic loss and affected population in China to 1.5°C and 2°C of global warming. The topic is very interesting. However, there are some points that are not clear to me:

**Response**: *We appreciate your critical comments and have incorporated your constructive points into our revision. We would like to outline our revision point by point in the following sections.*

Data: the reference of SSP-RCP combination scenarios should be given.

**Response:** *Thanks for the constructive comments made by the referee. To simulate climate change and assess its impact, mitigation and adaptation capabilities, IPCC provides corresponding information on RCPs and SSPs scenarios. Where, RCP8.5 scenario is generally comparable to the SSP3 or SSP4 scenarios; RCP6.0 scenario is generally comparable to the SSP2 scenario; and RCP4.5 scenario is generally comparable to the SSP1 scenario (Kriegler et al., 2010; Van Vuuren et al., 2012).*
*In the revised manuscript, Lines 76-77 has been revised to "The SSP1 and SSP3 scenarios were selected, which corresponding to RCP4.5 and RCP8.5 scenarios, respectively (Kriegler et al., 2010; Van Vuuren et al., 2012)".*
*References:*
*Kriegler, E., O'Neill, B. C., Hallegatte, S., Kram, T., Lempert, R., Moss, R. H., & Wilbanks, T. J. (2010). Socio-economic Scenario Development for Climate Change Analysis (No. 2010-23, pp. 3-3). CIRED Working Paper DT/WP.*
*Van Vuuren, D. P., Riahi, K., Moss, R., Edmonds, J., Thomson, A., Nakicenovic, N., ... & Rose, S. K. (2012). A proposal for a new scenario framework to support research and assessment in different climate research communities. Global Environmental Change, 22(1), 21-35.*

Methods: how the "I—environmental correction parameter" in Eq. 2 the "D—the destructive power" are calculated?

**Response:** *Thanks for the constructive comments made by the referee. The construction process of "I—environmental correction parameter" and "D—the destructive power" have been added in Lines 109-113 and Lines 118-122 of the revised manuscript.*

L111: Why 30 (35)? Any reason?

**Response:** *Thanks for the constructive comments made by the referee. In the study, we found that in a continuous rainfall process, the maximum accumulated 3-day precipitation that causes flood disasters is generally at least 30 mm, with a slight difference between north and south. In the south of the Yangtze River, more than 35 mm is generally required, while in the north of the Yangtze River, more than 30 mm is sufficient. In the revised manuscript, Lines 97-103 has been revised to "Generally, mild, moderate, and severe floods correspond to maximum accumulated 3-day precipitation values of 30(35)–150 mm, 150–250 mm, and ≥ 250 mm, respectively (Li et al., 2012). For flood disasters in the south of the Yangtze River, the maximum accumulated 3-day precipitation of more than 35 mm is generally required, whereas for flood disasters in the north of the Yangtze River, the maximum accumulated 3-day precipitation of more than 30 mm is sufficient. Therefore, for the classification of mild flood disasters, the maximum accumulated 3-day precipitation is at a minimum of 35 mm in the south of the Yangtze River and 30 mm in the north of the Yangtze River."*

Figure 2, the hazard of flood? I guess it is the probability of flood. Please make it clear.

**Response:** *Thanks for the constructive comments made by the referee. In this study, the hazard of flood was represented by the probability of the occurrence of the flood disaster, and we have amended the statement in the manuscript to avoid misinterpretation. In the revised manuscript, title of Figure 2 has been revised to "Spatial patterns of the probability of the severe (a-b), moderate (c-d), and mild (e-f) floods for 1.5°C (a, c, e) and 2°C (b, d, f) of global warming under RCP8.5 scenario".*

How Figure 7 is obtained? A combination of Figure 2 & Figure 4? Why the risks in the north part of Xinjiang are so high (here, the values in Figure 2 & Figure 4 are low), some explanations are needed. I strongly suggest to separating the affected population risk from the economic loss risk.

**Response:** *Thanks for the constructive comments made by the referee. The integrated flood risk is obtained by combining population and economic risks using the superposition analysis. Figure 7a is obtained by combining Figure S3a, c, e and Figure*

*S5a, c, e. Figure 7b is obtained by combining Figure S3b, d, f and Figure S5b, d, f. Figure 7c is obtained by combining Figure 4a, c, e and Figure 6a, c, e. Figure 7d is obtained by combining Figure 4b, d, f and Figure 6b, d, f. We redrew the figures in the manuscript, fixing some of the errors and corresponding descriptions. It can be seen in Figures 2,4,6,7 and the Results section of the revised manuscript and Figures S1-6 of the Supplementary Material.*

*In the Results section of the revised manuscript, we have separated population and economic risk into two parts. It can be seen in sections 3.2 and 3.3.*

The writing should be further improved, e.g.

"social and economic risks of the floods" is suggested to be changed to "the affected population and economic risks of the floods", e.g. in L11, L107.

**Response:** *Thanks for the constructive comments made by the referee. In the revised manuscript, changes have been made in accordance with the comments of the referee.*

"flood" should be "floods" in the title.

**Response:** *Thanks for the constructive comments made by the referee. Based on the opinions of reviewers and Accdon-LetPub editors, the title was revised to "Half a degree of warming may cause double the economic loss and increase the population affected by floods in China".*

Lines 9-19, correct some grammatical errors in Abstract, such as "analyze" should be "analyzed".

**Response:** *Thanks for the constructive comments made by the referee. The manuscript has been edited by Accdon-LetPub to correct the errors.*

Line129, "km2" should be "km$^2$".

**Response:** *Thanks for the constructive comments made by the referee. Similar errors in the manuscript have been corrected.*

The manuscript contains a few grammatical errors that require English editing services for language correction.

**Response:** *Thanks for the constructive comments made by the referee. The manuscript has beenedited by Accdon-LetPub, and we have corrected grammatical errors in the manuscript.*

Review 2:

This paper analyzes the population and economic risks of floods under different global warming periods using future climate and socioeconomic scenarios data, and obtains some meaningful conclusions, which has certain guiding significance for the formulation of disaster prevention and mitigation measures. The whole article is good, so I tend to recommend the paper be accepted by the journal after minor revisions. The specific suggestions are as follows:

**Response**: *We have incorporated your critical comments in our revised paper accordingly. In the space below, we respond point to point to each of your specific comments*

1. I think the most issue of this manuscript is the English language usage. Many sentences in the manuscript cannot be understand accurately and even some grammatical errors exist in the manuscript.

**Response:** *Thanks for the constructive comments made by the referee. The manuscript has been polished by Accdon-LetPub, and we have corrected grammatical errors in the manuscript.*

2. The usages of "flood", "floods" and "flooding" in the manuscript need to be unified.

**Response:** *Thanks for the constructive comments made by the referee. With the help of Accdon-LetPub, we have unified the usage of the words in the manuscript.*

3. The title of 2.2.1 is inaccurate and needs to be revised.

**Response:** *Thanks for the constructive comments made by the referee. The title of 2.2.1 has been revised to "Determination of 1.5°C and 2°C global warming periods".*

4. Authors need to distinguish the difference between "hazard" and "risk".

**Response:** *Thanks for the constructive comments made by the referee. We have clarified the description of risks and hazards in the manuscript, and we have revised the title of Section 3.1 and the title of Figure 2 to avoid misunderstanding.*

5. Figure 2,4,6 in the manuscript is not very clear and I suggest redrawing it.

**Response:** *Thanks for the constructive comments made by the referee. We have redrawn the figures in the manuscript, fixing some of the errors and corresponding descriptions.*

6. The pictures in the supplemental material have the same problem.

**Response:** *Thanks for the constructive comments made by the referee. We have redrawn the figures in the supplemental material.*

7. Line 81, "warming targets" should be changed to "warming periods". Applies to other places. And the sentence needs to be modified.

**Response:** *Thanks for the constructive comments made by the referee. In the revised manuscript, Line 81 has been revised to "We selected global warming periods of 1.5°C and 2°C based on the objectives of the Paris Agreement". Other descriptions in the manuscript have been revised accordingly.*

8. The conclusion section needs to be reorganized based on the study of the manuscript. For example, the first sentence of Line 285 is not sufficient for a conclusion.

**Response:** *Thanks for the constructive comments made by the referee. We have reorganized the conclusions section. In the revised manuscript, Line 319 has been revised to "Floods mainly affect the population and economy of eastern China.".*

---

## Author Response (AR2)

The authors present an interesting work on the assessment of flood risk under different warming period. The work is well designed and relevant to the theme of the special issue. After an anonymous review, the changes made in the original manuscript and the responses to comments have improved the manuscript considerably. It is therefore recommended for publication. However, several aspects need to be addressed and several points clarified in order to continue to improve the quality of the article. I summarize my comments below:

**Response**: *Thanks for your advice. You provided good suggestions to help us improve*

the research and explain our findings. We would like to outline our revision point by point in the following sections.

1. Introduction

• Lines 24-26, the high incidence and severity of floods demonstrates the need for research. Damage data can be increased and updated based on the latest reports.

**Response**: Thanks for the constructive comments made by the referee. We have updated the data based on the latest reports published by international organizations. In the revised manuscript, Lines 24-26 has been revised to "Between 2000 and 2019, 7348 disaster events were recorded worldwide, resulting in 1.23 million deaths, \$2.97 trillion USD in economic losses and over 4 billion people affected, with a surge in the number of climate-related disasters, of which floods were the most frequent, accounting for 44% of all disasters. (CRED and UNDRR, 2020; WMO, 2021)."

Reference:

WMO: Atlas of Mortality and Economic Losses from Weather, Climate and Water Extremes (1970-2019), WMO, https://library.wmo.int/index.php?lvl=notice\_display& id=21930#.Yj6D3-dBxI0., last access: 26 March 2022, 2021.

CRED, and UNDRR: The human cost of disasters: an overview of the last 20 years (2000-2019), https://www.undrr.org/publication/human-cost-disasters-overview-last-20-years -2000-2019., last access: 26 March 2022, 2020.

• Lines 31-32, same as above. It is recommended to update the data and

references.

**Response**: Thanks for the constructive comments made by the referee. We have updated the data based on the latest statistical report from the China Meteorological Administration. Lines 31-32 has been revised to "Over the last ten years, the mean annual direct economic loss due to floods in China has exceeded \$25 billion USD, and the affected population has exceeded 100 million with more than 1000 deaths (Song, 2019; Chen et al., 2022)."

**Reference:**

Chen, Y., Wang, L., Zhao, J., Zhang, Y., Zhao, S., Li, W., Zou, X., Jiang, Y., Shi, S., Hong, J., Li, D., Wang, Y., Hou, W., Zhu, X., Dai, T., Cai, W., Guo, Y., Zhong, H., and Wang, Q.: Climatic Characteristics and Major Meteorological Events over China in 2021, Meteorological Monthly, https://doi.org/10.7519/j.issn.1000-0526.2022.022501, 2022. Song L: China meteorological disaster yearbook 2019: Beijing, China, China Meteorological Press, China, 2019.

• The IPCC Sixth Assessment Report has been released. It is suggested to add relevant content to the second paragraph.

**Response**: Thanks for the constructive comments made by the referee. In the revised manuscript, the key findings of the IPCC Sixth Assessment Report have been added to the revised draft, namely: "The IPCC's Sixth Assessment Report indicates that global surface temperature was 1.09°C higher in 2011–2020 than 1850–1900 and that global warming will reach 1.5°C in the near-term, which will cause unavoidable increases in multiple climate hazards and present multiple risks to ecosystems and humans (IPCC, 2021; IPCC, 2022). "

*Reference:*

*IPCC: Climate change 2021: the physical science basis, Cambridge University Press, UK, 2021.*

*IPCC: Climate change 2022–Impacts, adaptation and vulnerability, Cambridge University Press, UK, 2022.*

2. Material and Methods

• Lines 67, why choose RCP4.5 and RCP8.5 scenarios?

**Response**: Thanks for the constructive comments made by the referee. In the revised manuscript, we have added the reasons for choosing the RCP4.5 and RCP8.5 scenarios in Section 2. Materials and Methods.

• Lines 129-133, add figures of destructive power of floods.

**Response**: Thanks for the constructive comments made by the referee. In the revised manuscript, we have shown the destructive power of different levels of floods in Figure *S1* of supplementary materials.

• Line 134, adding descriptions and figures, combined with the previous suggestion can help the reader understand the content.

**Response**: Thanks for the constructive comments made by the referee. In the revised manuscript, lines 141-143 has been revised to "The exposures of the social economy were obtained from the SSPs data described in the data sources section. The annual population and GDP data were obtained based on linear interpolation method, and then the average values of population and GDP at different global warming periods under different scenarios were calculated to represent the exposure (Fig.S2; Fig. S3)".

3. Results

• Lines 148-149, notes on figure names should follow the text rather than being stacked together. Same below.

**Response**: *Thanks for the constructive comments made by the referee. In the revised manuscript, notes on figures follow the text to make it more readable.*

• Figure 2, The colors in the picture are difficult to distinguish. It should look like Figures 4 and 6 below.

**Response**: Thanks for the constructive comments made by the referee. In the revised manuscript, we have redrawn Figures 2 and S4 with different colors to make them legible.

- 4. Discussion
- Lines 279-281, according to Equation (3), the last element is the

environmental correction parameter, which is 0.5552 and 0.5640 respectively. Line282 mentions that "Because the vulnerability and the environmental correction parameter would be basically unchanged" is incorrect. Am I understanding this correctly?

**Response**: Thanks for the constructive comments made by the referee. Sorry for the mistake. The statement " the vulnerability and the environmental correction parameter would be basically unchanged " is correct. The environmental correction parameter in Lines 286-291 should be 0.5552. In the revised manuscript, this part has been amended to "According to Equation (3), for  $1.5^{\circ}$ C of global warming, the population affected (units: million) and the direct economic losses (units: US\$ billion) caused by severe floods in Northeast China would be  $0.2880 \approx 30.73\% \times 129.8462 \times 0.0130 \times 0.5552$  and  $1.0518 \approx 11.83\% \times 1231.8067 \times 0.0130 \times 0.5552$ , respectively. For  $2^{\circ}$ C of global warming, the population affected and the direct economic losses caused by severe floods would be  $0.3615 \approx 30.73\% \times 145.1178 \times 0.0146 \times 0.5552$  and  $2.4794 \approx 11.83\% \times 2585.6269 \times 0.0146 \times 0.5552$ , respectively."